# Unusual Association of NF-κB Components in Tumor-Associated Macrophages (TAMs) Promotes HSPG2-Mediated Immune-Escaping Mechanism in Breast Cancer

**DOI:** 10.3390/ijms23147902

**Published:** 2022-07-18

**Authors:** Veronica De Paolis, Fabio Maiullari, Maila Chirivì, Marika Milan, Chiara Cordiglieri, Francesca Pagano, Alessandra Rita La Manna, Elena De Falco, Claudia Bearzi, Roberto Rizzi, Chiara Parisi

**Affiliations:** 1Institute of Biochemistry and Cell Biology, National Research Council of Italy (IBBC-CNR), Via Ercole Ramarini, 32, Monterotondo, 00015 Rome, Italy; veronica.depaolis@ibbc.cnr.it (V.D.P.); francesca.pagano@cnr.it (F.P.); 2Fondazione Istituto Nazionale di Genetica Molecolare, Via F. Sforza, 35, 20122 Milan, Italy; maiullari@ingm.org (F.M.); marika.milan@policlinico.mi.it (M.M.); cordiglieri@ingm.org (C.C.); claudia.bearzi@itb.cnr.it (C.B.); 3PhD Program in Cellular and Molecular Biology, Department of Biology, University of Rome “Tor Vergata”, Via della Ricerca Scientifica, 1, 00133 Rome, Italy; 4Department of Molecular Medicine Sapienza University, Viale Regina Elena, 324, 00161 Rome, Italy; maila.chirivi@uniroma1.it; 5UOC Neurology, Fondazione Ca’Granda, Ospedale Maggiore Policlinico, Via F. Sforza, 28, 20122 Milan, Italy; 6Breast Unit, Department of General Surgery, Santa Maria Goretti Hospital, Via Guido Reni Snc, Azienda Unità Sanitaria Locale (AUSL), 04100 Latina, Italy; al.lamanna@ausl.latina.it; 7Department of Medical Surgical Sciences and Biotechnologies, Sapienza University of Rome, C.so della Repubblica, 79, 04100 Latina, Italy; elena.defalco@uniroma1.it; 8Clinica Mediterranea Cardiocentro, Via Orazio, 2, 80122 Napoli, Italy; 9Institute for Biomedical Technologies, National Research Council, Via Fratelli Cervi, 93, 20054 Segrate, Italy

**Keywords:** tumor-associated macrophages, HSPG2, NF-κB, 3D culture

## Abstract

The cellular heterogeneity of the tumor environment of breast cancer (BC) is extremely complex and includes different actors such as neoplastic, stromal, and immunosuppressive cells, which contribute to the chemical and mechanical modification of the environment surrounding the tumor-exasperating immune-escaping mechanisms. In addition to molecular signals that make the tumor microenvironment (TME) unacceptable for the penetrance of the immune system, the physical properties of tumoral extracellular matrix (tECM) also have carved out a fundamental role in the processes of the protection of the tumor niche. Tumor-associated macrophages (TAMs), with an M2 immunosuppressive phenotype, are important determinants for the establishment of a tumor phenotype excluded from T cells. NF-κB transcription factors orchestrate innate immunity and represent the common thread between inflammation and cancer. Many studies have focused on canonical activation of NF-κB; however, activation of non-canonical signaling predicts poor survival and resistance to therapy. In this scenario, we demonstrated the existence of an unusual association of NF-κB components in TAMs that determines the deposition of HSPG2 that affects the stiffness of tECM. These results highlight a new mechanism counterbalanced between physical factors and a new perspective of mechano-pathology to be targeted to counteract immune evasion in BC.

## 1. Introduction

Breast cancer (BC) is the most common cancer in women [1]. Refined clinical stratification is based on distinctive molecular expressions that can predict prognosis and strategically delineate personalized clinical intervention [2].

Specifically, the presence on the surface of neoplastic cells of the estrogen receptor (ER) and the progesterone receptor (PR), together with the expression and amplification of the human epidermal growth factor type 2 (HER2) receptor, allows for the classification of BC into three main clinical subtypes, namely, ER/PR positive, HER2 positive, or triple negative (TNBC) [3]. TNBC is the most aggressive subtype, and treatment options for metastatic TNBC are still limited [4]. Novel and recent insights on the biological hallmark of cancer have suggested a key role of the complex interaction between cancer cells and the tumor microenvironment (TME). These interactions have only recently emerged thanks to the development of innovative technologies for the faithful recapitulation of physiological intersystem relationships in vitro.

BC is characterized by a peculiar TME where tumor-infiltrating lymphocytes, heterogeneous subtypes of myeloid cells, cancer-associated fibroblasts (CAFs), and tumor-associated macrophages (TAMs) orchestrate and redefine the composition of the tumor ECM (tECM) [5,6].

Recent research has highlighted the significant impact of the immune component on TME remodeling [7].

Specifically, TAMs are the main immune components of BC TME and are characterized by a distinct immunosuppressive function raising tumor progression. Despite this, the contribution of TAMs in conditioning ECM in a pathological system is still unexplored.

Generally, the massive remodeling of the tECM is carried out by CAFs, but recent discoveries have shown that the continuous crosstalk with the other actors of the TME contributes to define the matrix molecular architecture, the physical structure, and mechanical properties [8].

Proteoglycans are fundamental components of the ECM, and their suppressive or agonist modulation triggers molecular signals capable of modulating the behavior of cell populations included in the physiological microenvironment [9].

Among these, heparan sulfate proteoglycan 2 (HSPG2), also known as perlecan, is massively deposited in the TME of invasive and metastatic carcinomas and various tumors, more generally all those undergoing epithelial–mesenchymal transition (EMT) [10]. HSPG2 is a 470 kDa core protein bound to three heparan sulfate (HS) glycosaminoglycan chains. In TNBC, it is detected both in primary tumor and in metastatic lesions. Its high levels have been correlated with poor survival and poor prognosis [11]. Indeed, HSPG2 is involved in drug resistance in BC in the hormone receptor positive subtype [12]. However, the cellular and molecular mechanisms leading to the deposition of HSPG2 within BC TME have not been studied.

The nefarious correlation with the activation of transcription factors, distinctive for neoplastic transformation, such as the heterodimeric complex nuclear factor-ĸB (NF-ĸB) RelA (p65)/p50 [13], led us to pursue an in-depth investigation of the possible pathological relationship with the molecular actors of TME, and in particular with the proteoglycan HSPG2.

The p65 subunit has been already implicated in the transcriptional regulation of ECM components in solid tumors, and the transcriptional activation of p65 increases HSPG2 expression in the desmoplastic prostate TME [14].

The NF-κB family is characterized by sequence homology in the DNA-binding sequence and dimerization domains. These features allow binding to similar DNA response elements in the genome, and participation in similar biological programs, through transcriptional activation and repression of hundreds of genes. Despite the close relationship between the NF-κB family members, they all show distinct activities [15].

In particular, the transcriptional program of NF-ĸB is strictly regulated by the composition of its active forms, which are homo- and heterodimers consisting of five monomers (p50, p52, p65, c-Rel, and RelB).

A typical factor is a heterodimer involving p50 or p52, which are limited processing products of p105 or p100, respectively, and a member of the Rel protein family, typically p65. The most common heterodimers are p65/p50, which are activated through the canonical pathway, and RelB/p52 also known as the non-canonical mode.

Despite being biologically feasible and to some extent observed [16,17,18,19], the existence of the p65/p52 association and its role in TME cancer has never been appreciated.

Recent scientific literature corroborated the fundamental advances obtained in the oncology field, thanks to the state of the art of technological development, which can now be used by experimental investigation [20]. In particular, the development of biomaterials has allowed not only the three-dimensional recapitulation of the tumor system, but also the possibility of studying multifactorial interactions linked to the biological, physical, and mechanical system.

Our study used modulable biomaterials that can connect different TME actors, also highlighting the mechanical properties of the matrix, which have shown to have a fundamental role in the modulation of gene expression at the level of different cell populations and in detail of the immune system.

## 2. Results

### 2.1. HSPG2 Overexpression in TNBC by TAM

We performed immunofluorescence analysis on tissue sections of patients with TNBC to verify the expression of HSPG2 within the TME. Our results showed that the expression of HSPG2 was enhanced and highly localized in the stromal elements of the neoplastic area (Figure 1a) compared to the surrounding non-neoplastic region (Appendix A).

The corresponding sections of TNBC were also labelled with a panel of markers including CD206 and wheat germ agglutinin (WGA), employed to mark resident activated macrophages exhibiting M2 phenotype and membrane-specific glycoproteins, respectively. The images (Figure 1c) illustrate the colocalization of HSPG2 and CD206, suggesting that HSPG2 was mainly expressed in the peritumoral tissue and likely ascribable to TAM production. Notably, CD206^+^ macrophages were highly infiltrated in the neoplastic area and distributed in its proximity, in accordance with HSPG2 deposition, while in the non-neoplastic surrounding area were hardly detected (Figure 1b). Together, these results indicate that the high infiltration of TAM in the TNBC TME may contribute to the abnormal deposition of stromal HSPG2.

### 2.2. HSPG2 Transcriptional Modulation and Unconventional NF-κB Balance in M2 Macrophages

We performed a 2D model with THP-1 cell line differentiated into Mϕ by phorbol-12-myristate-13-acetate (PMA) administration in order to deeply characterize changes of HSPG2 expression in human macrophages associated with BC subtypes. Mϕ were stimulated with LPS or IL-4 to induce M1 or M2 polarization, then untreated Mϕ and M2 were concomitantly exposed to tumoral conditioned medium (TCM) from hormone-responsive MCF-7 (epithelial model of metastatic adenocarcinoma of BC) or triple-negative MDA-MB-231 BC cell lines for in vitro TAMs induction [21]. 

Gene expression analysis was performed to validate the occurred polarization (Figure 2a). M2 macrophages, as expected, showed an enhanced expression of CD206, CCL22, and TGM2 genes, correlated with lower levels of CXCL9, a marker of M1 polarization. 

Accordingly with the recent literature, both Mϕ and M2 TAMs showed a marked-up regulation of CD206, CCL22, and TGM2 M2 genes. Among them, the CCL22 chemokine has been related to the immunosuppressive action exerted by TAMs in the TME of different types of cancer [22].

Surprisingly, our data showed significant higher levels of both CD206 and CCL22 M2 together with CXCL9 M1 gene enhancement in MCF7 compared to MDA-MB-231 associated TAMs, thus highlighting the existence of specific macrophage behavior when exposed to factors released by different BC subtypes.

To further characterize TAMs polarization, we performed immunofluorescence analysis of CD163, which is a phenotypic membrane marker of M2 macrophages. All induced TAMs showed a marked CD163 labelling associated with M2 typical elongated shape, as reported in Figure 2b.

Afterward, we assessed HSPG2 and NF-κB pathway gene expression changes in macrophages subtypes (Figure 3a). A mild trend of HSPG2 induction was observed in M2 macrophages, consistent with previous reports [23].

We observed a significant over-expression of HSPG2 mRNA in all TAMs with a higher induction when M2 TAMs were associated with MCF-7 TCM compared to the association with MDA-MB-231 TCM (4.92- vs. 2.82-fold, *p* = 0.0043). 

We then confirmed a marked HSPG2 deposition, with positive membrane-bound labelling in all TAM populations by immunofluorescence analysis, consequently to an HSPG2 mRNA upregulation (Figure 3b). 

Our results confirmed that HSPG2 is produced by TAMs and that its overexpression is additionally modulated by metabolic changes [24,25] that characterize different cancer cell lines.

Moreover, we observed the RelA gene overexpression in all TAMs together with a significant enhanced NFKB2 gene induction.

In the canonical NF-κB activation pathway, the heterodimeric RelA/p65-NFKB1 complex appeared to be the most abundant, while in the non-canonical one, the NFKB2/p100 phosphorylation and processing induced the formation of the heterodimeric RelB-p52 complex [15]. Surprisingly, both RelB and NFKB1 showed no significant changes in macrophages, suggesting that a specific activation of NFKB2 and RelA could promote the formation of an unusual heterodimeric p65/p52 complex, correlated to HSPG2 transcriptional regulation.

### 2.3. NF-κB Was Responsible for HSPG2 Expression in TAMs, Pointing to the Unexplored p65/p52 Heterodimer Formation

We then evaluated whether the NF-κB pathway could be responsible for HSPG2 transcriptional changes in TAMs by the administration of 6-amino-4-(4-phenoxyphenylethylamino) quinazoline (QNZ), a specific NF-κB inhibitor. Gene expression analysis (Figure 4a) demonstrated a significant reduction of HSPG2 gene expression in TAMs (0.64-fold in MϕTAM^MDA-MB−231 TCM^; 0.37-fold in MϕTAM^MCF−7 TCM^, *p* < 0.033), together with down-modulation of the common NF-κB target genes IL-6, IL-1b, RelA, and NFKB2, thus validating HSPG2 as a novel target of NF-κB pathway activity, speculating a directed regulation by unusual p65/p52 association.

In resting condition, inactivated p65 and p100 precursor proteins are sequestered in the cytosol, while proper NF-κB activation leads to p65 release or p52 processing and release with consequent nuclear translocation, leading to gene regulation of several target genes. 

We thus investigated if the observed p65 and p52 gene upregulation corresponded to their activation by quantifying protein distribution in the cytoplasmic and nuclear fractions. 

As shown in Figure 4b, both p65 (38.38% in MϕTAM^MDA-MB−231 TCM^, *p* = 0.029, and 58.65% in MϕTAM^MCF−7 TCM^, *p* = 0.002, vs. 11.07% in M2) and p52 (33.60% in MϕTAM^MDA-MB−231 TCM^ and 38.75% in MϕTAM^MCF−7 TCM^ vs. 14.14% in M2, *p* < 0.03) proteins were enriched in the nuclear fraction of TAMs, indicating their concomitant activation. We examined endogenous interaction of p65 and p52 in both Mϕ macrophages and TAMs nuclear extracts. Co-immunoprecipitation (co-IP) experiments, reported in Figure 4c, demonstrated that p65 antibody specifically co-precipitated endogenous p52 (Figure 4c, lower panel), revealing the heterodimerization between p65 and p52 in both Mϕ and TAMs. These data validate the existence of an unexpected heterodimeric p65/p52 complex in macrophages.

### 2.4. HSPG2 Was Expressed on the Surface of M2 Macrophages in the t3D Model

Within the TME, the mechanical rigidity caused by a dysregulated ECM has been correlated to a worse prognosis and failure of pharmacological therapies [26]. To profile this mechano-pathological perturbation, we generated an in vitro cancer 3D model to recapitulate the physical properties of the tECM in order to retrace the molecular pathways involved in the expression of HSPG2 by M2 macrophages.

Poly (ethylene glycol)-fibrinogen (PEG-FB) hydrogels were used, as referred by Chirivì et al. (2021), to mimic the stiffness of healthy breast tissue (h3D) and the five times stiffer matrix of triple-negative breast cancer (t3D) [27].

THP-1, alone or in association with MCF-7 or MDA-MB-231 cell lines, were encapsulated in the h3D and t3D matrix and polarized into M1 or M2 phenotypes.

Immunofluorescence analysis of CD206 and HSPG2 (Figure 5a), performed in Mϕ and M2 macrophages, showed that the t3D matrix successfully induced a high level of CD206^+^ M2 macrophages. Intriguingly, CD206^+^ M2 macrophages directly correlated with the increase in HSPG2 expression present in the surrounding stroma (Figure 5b), a peculiar feature of TAMs.

### 2.5. Cancer Stiffness Exacerbated HSPG2/NF-κB Dysregulation in the TNBC Model

The high expression of TGM2, a marker of M2-activated macrophages, together with the positive expression of the CD206 marker and the low level of CXCL9, demonstrated, for the first time, the correct M2 polarization in our 3D modulable system (Figure 6a). In particular, as shown in Figure 6b, samples included in the t3D matrix showed more significant transcriptional changes than healthy ones. We observed that in the tumor matrix, the expression of HSPG2 was increased in M2 macrophages co-assembled with MDA-MB-231 cells compared to M2 + MCF7 cells (1.72- vs. 0.34-fold, *p* = 0.0008), correlated with the deposition of HSPG2 by CD206^+^ macrophages in TNBC biopsies (Figure 1c). On the other hand, in the healthy matrix, HSPG2 was increased in M2 macrophages but not in the presence of tumor cells (1.68- vs. 0.33-fold in M2 + MCF7 cells, *p* = 0.0037), suggesting that neoplastic cells and correct matrix stiffness are both required to induce HSPG2 expression. In M2 macrophages embedded together with MDA-MB-231 cells in the t3D matrix, we observed that NFKB2 (2.70- vs. 1.29-fold, *p* = 0.01) and RELA (2.61- vs. 0.97-fold, *p* = 0.001) was significantly upregulated compared to M2 alone, while there were no significant changes in NFKB2 (0.57- vs. 1.29-fold) and RELA (0.37- vs. 0.97-fold) expression in M2 macrophages + MCF-7 cells (Figure 6c).

These results suggest that in a 3D system, high-fidelity mimicking the mechano-pathological conditions of TNBC TME, M2 macrophages encapsulated with triple-negative tumor cells exhibited specific expression of HSPG2. Furthermore, the correct differentiation of M2 associated with a stiffer matrix significantly modulated the expression of genes of the NF-κB pathway, confirming the involvement of heterodimeric p65/p52 complex in the transcriptional regulation of HSPG2 and in a tumor immune-escaping process.

## 3. Discussion

Infiltration of CD8^+^ T cells into TME plays a key role in cancer immune surveillance limiting neoplastic progression [28]. In fact, the absence of CD8^+^ T cells in TME of many solid tumors, as well as the high density of TAMs, is associated with poor prognosis [29].

Activation of TAMs promotes tumor growth and progression through several mechanisms, including the production of pro-tumor growth factors, which stimulate tumor angiogenesis, matrix remodeling, tumor cell migration, and the promotion of immunosuppression [30].

Many mechanisms are activated by the actors belonging to the TME to modulate the immune action, starting from the secretion of factors with immunosuppressive properties [31,32,33], up to a massive mechanical remodeling of the peritumoral ECM, which inhibits the migration of T cells.

The proteoglycan HSPG2, one of the most represented molecules of tECM of various types of cancer [34], actively contributes to support tumor growth by sequestering proteins, growth factors, cytokines, and enzymes through its heparan sulfate (HS) chains [35], generating gradients that control the development of cancer cells.

The manipulation of the TME represents a promising clinical strategy and will pave the way for “precision immunotherapy” in cancer patients. 

In a recent TNBC study, an anti-HSPG2 antibody successfully inhibited tumor growth in a mouse xenograft model [11], highlighting the promising role of HSPG2 as a therapeutic target in tumor cells.

To date, most of the research has focused on the production of HSPG2 by cancer cells [36], while, only recently, the fundamental role of additional TME actors in its pathological genesis has emerged [37,38].

In pancreatic cancer, tumor cells epigenetically reprogram CAFs that promote HSPG2 release through an NF-κB-dependent pathway, increasing invasion, metastasis, and chemoresistance to gemcitabine [39]. Furthermore, in a TAM-deficient colorectal tumor model, several ECM-related proteins, including HSPG2, were downregulated, suggesting the contribution of TAMs in HSPG2 deposition [40].

However, very little is known in TNBC about TME actors involved in HSPG2 deposition and the molecular mechanisms that contribute to HSPG2 dysregulation, including its transcriptional regulation.

In this study, for the first time, we demonstrated that TNBC tissue exhibits a distinctive stromal deposition of HSPG2 in high cell density areas and surrounding tumor islands where TAMs are highly infiltrated, mimicking a physical shield. The positive correlation of TAMs (CD206^+^ macrophages) with HSPG2 suggests their role in increasing the stiffness of the tECM through the deposition of HSPG2. We hypothesized that the defined architecture of the peri-tumoral ECM could be a significant TAMs-mediated immune-escape mechanism.

We therefore explored the expression profile of the HSPG2 and NF-κB pathway genes. 

Our data demonstrated that the conditioned medium of tumor cells significantly induces polarization of the M2 sub-phenotype, leading to an upregulation of M2 markers. Surprisingly, HSPG2 mRNA was significantly upregulated in macrophages exposed to TCM, suggesting that secretome released by tumor cells may contribute to exacerbating TAM activation and HSPG2 production.

Furthermore, we observed that the overexpression of HSPG2 occurs concomitantly to enhanced expression of the components of the canonical and non-canonical signaling pathways RelA (p65) and NFKB2 (p52). In accordance, TAMs treated with the NF-κB inhibitor QNZ showed significant downregulation of HSPG2 mRNA, demonstrating HSPG2 as a novel target of NF-κB pathway activity.

Intriguingly, our data showed the existence of endogenous p65/p52 interaction in nuclear extracts of macrophages and TAMs, suggesting that the dimer is active and ready to bind DNA elements that regulate the transcription of target genes.

The interaction of TAMs with cells in 3D culture mimics the mechano-pathological tumor condition, which evidently is not appreciated in 2D models. In fact, we demonstrated enhanced CD206 expression and a peculiar distribution of HSPG2, showing clusters of protein within cell cytoplasm and in the cell surrounding area, reminiscent of protein secretion, in M2 macrophages encapsulated in the t3D system, mimicking what has been observed in TNBC biopsies. Additionally, the correct spatial distribution among TAMs and tumor cells within the cancer matrix confirmed p65/p52 gene activation in BC, revealing an exacerbated overexpression specifically in TNBC subtype, in accordance with what observed in vivo, thus confirming its involvement in tECM remodeling. In recent years, it has emerged that the complexity of TME is exacerbated by the pathological rigidity that characterizes tECM, underlining the significant importance of mechano-pathology in the study of cancer progression [41].

Our results revealed how the progress of oncological therapies can no longer be separated from a multifactorial approach that takes into consideration the different actors of TME, not underestimating the support of physical and mechanical factors for tumor progression.

The development of more advanced 3D bio-assembly technologies has made it possible to faithfully recapitulate pathological systems in vitro, and overwhelming results, until now, are considered milestones [42].

## 4. Materials and Methods

### 4.1. Cell Lines

THP-1 human monocytic cells, MCF-7 human breast cancer cells, and MDA-MB-231 human triple-negative breast cancer cells were obtained from ATCC and maintained in RPMI-1640 (Sigma-Aldrich, St. Louis, MO, USA) medium supplemented with 10% fetal bovine serum (FBS) (Gibco, Waltham, MA, USA), 100 U/mL penicillin (Gibco), and 100 mg/mL streptomycin (Gibco) at 37 °C with 5% CO2. To obtain a tumor-conditioned medium (TCM) of MDA-MB-231 and MCF-7 cancer cells, approximately 2 × 10^5^ cells per mL were grown to 80% of confluence, and the supernatants were harvested and centrifuged for 5 min at 500× *g* to remove suspended cells, then filtered (70 μm, Corning, Glendale, Arizona, USA) and collected.

For differentiation toward a macrophage phenotype, THP-1 cells were plated at a density of 3 × 10^5^ cells per mL, or 1.5 × 10^5^ cells per mL for microscopy analysis, and incubated with 100 ng/mL of the phorbol 12-myristate 12-acetate (PMA) (P8139; Sigma-Aldrich) diluted in media. After 24 h, PMA-containing media was replaced with fresh media, and adherent THP-1 cells were left in culture for 72 h before treatments. Macrophages were maintained in Mφ state, or they were M1 or M2-activated with LPS (100 ng/mL, L4516; Sigma-Aldrich), IL-4 (20 ng/mL, 130-093-921; Miltenyi Biotec, Bergisch Gladbach, Germany), or with the addition of TCM (half of the total plated volume) for 24 h or 48 h. For the NF-κB inhibitor treatment, Mφ macrophages were pre-activated to M2-phenotype with a tumor-conditioned medium for 16 h and then treated with 50 nM of 6-amino-4-(4-phenoxyphenylethylamino)quinazoline (QNZ) (EVP4593, 481406; Calbiochem, San Diego, CA, USA) or dimethyl sulfoxide (DMSO) (Sigma-Aldrich) for 48 h. 

### 4.2. Hydrogel Formulation

The 3D bulk constructs were generated using poly (ethylene glycol)-fibrinogen (PEG-FB) with a G’ = 385 Pa in a stock concentration of 11 mg/mL. The decision to use PEG-FB was motivated by its peculiarity in modulating its biochemical and mechanical characteristics and to exclude tumor variability due to different proportions of various collagen types. The healthy tissue (h3D, Young’s modulus <500 Pa [27]) was simulated using the PEG-FB at 5 mg/mL concentration in PBS, while the stiffer cancerous tissue (t3D, Young’s modulus ≥2500 Pa [27]) was mimicked using 8 mg/mL PEG-FB supplemented with 1% *w*/*v* polyethylene glycol diacrylate (PEG-DA) diluted in PBS. Irgacure 2959 was used as radical photoinitiator at a concentration of 0.01% *w*/*v*. THP-1 cells alone or in combination with MDA-MB-231 or MCF-7 cells were resuspended in each sterile PEG-FB working solution (5 mg/mL and 8 mg/mL + 1% PEG-DA) at a final concentration of 4 × 10^6^ cells/mL, poured into a 150 μL polydimethylsiloxane (PDMS) mold and cross-linked with UV light (365 nm, 4–5 mW/cm^2^) for 5 min. Finally, the obtained bulk samples were divided into different experimental groups and THP-1 were differentiated in Mφ, M1, and M2^IL−4^ macrophages. At the end of the established time point (96 h), the samples were processed to perform gene expression and immunofluorescence assays.

### 4.3. Gene Expression Analysis

Total RNA was extracted from macrophages of each experimental condition.

Macrophages encapsulated in the h3D and t3D samples were extracted from the bulks as previously described [27]. Briefly, 5 mg/mL PF and 8 mg/mL PF + 1% PEG-DA hydrogels were washed in PBS and finely minced. Then, the samples were treated with a digestion solution composed by 300 U/mL Collagenase II (337 U/mg; Worthington, Columbus, OH, USA) and 0.65 U/mL Collagenase D (0.29 U/mg; Sigma-Aldrich) in PBS (Euroclone, Pero, Milan, Italy) for 10 min (5 mg/mL PF bulks) and 25 min (8 mg/mL PF + 1% PEG-DA bulks) at 37 °C, and the digestion process was blocked with a solution of PBS-5% FBS. The cells were centrifuged at 1600 rpm for 5 min at RT and the collected cells were resuspended in 1 mL of TRizol (Invitrogen, Carlsbad, CA, USA) and homogenized with the TissueRuptor (Qiagen, Hilden, Germany). Differentiated macrophages cultured in 2D were resuspended in 0.5 mL of TRizol. 

RNA was extracted according to the manufacturer’s protocol, and RNA quantity was determined with the Nanodrop 100 System (Rockford, IL, USA). A total of 1 μg of total RNA was reverse transcribed to cDNA using the High-Capacity cDNA Reverse Transcription Kit (Applied Biosystem, Foster City, CA, USA). cDNA was used as a template to measure the expression of human CD206, CCL22, TGM2, CXCL9, HSPG2, RelA, NFKB2, NFKB1, RelB, IL-6, IL-1b, and GAPDH genes (Table 1) by quantitative real-time PCR (qRT-PCR) with the SensiFAST SYBR Hi-ROX Kit (Bioline, Meridian Bioscience, Cincinnati, OH, USA) using a 7900HT Fast Real-time PCR System equipped with SDS software (Applied Biosystems). The fold change of each target gene compared to the control Mφ macrophages was evaluated by ΔΔCt relative expression analysis.

### 4.4. Patients and Specimens

Cancer tissue samples were obtained during quadrantectomy from patients who underwent breast surgery for invasive ductal carcinoma. Samples were included in paraffin to diagnose and stage the cancer by histopathological analysis. The triple-negative phenotype (steroid hormone receptor absent, negative HER2 status) was confirmed by immunohistochemistry and FISH evaluation, and additional paraffin-embedded sections tissue sections were collected on microscopic slides, stained for hematoxylin and eosin, and visualized by optical microscopy to select the tumor area. 

This study was approved by the ethical committee of S. Maria Goretti Hospital (Latina, Italy; Approval Code: PTDA0; Approval Date: 23 October 2020), and informed consent was obtained in accordance with the Declaration of Helsinki of the World Medical Association. 

### 4.5. Immunofluorescence Analysis

Immunofluorescence analysis was performed on the tissue paraffin-embedded section, 2D culture, and t3D samples, following different protocols.

Paraffin-embedded tissue sections of invasive ductal carcinomas of the breast with the triple-negative phenotype (steroid hormone receptor absent, negative HER2 status evaluated by the hystopathological analysis) were collected on microscopic slides, which were stained for hematoxylin and eosin and visualized by optical microscopy to select the tumor area. Serial paraffin-embedded tissue sections were deparaffinized at 70 °C for 30 min, dewaxed with xylene, and dehydrated through descending alcoholic solutions. After washing in distilled water, the sections were pretreated in sodium citrate buffer pH 6.0 for 10 min in a microwave for the antigen-retrieval procedure. Blocking of non-specific binding was performed incubated section in 5% BSA-PBS solution for 1 h at room temperature (RT). Primary human antibodies were diluted 1:100 (HSPG2, MABT12; Millipore Sigma, Burlington, MA, USA) and 1:50 (CD206, ab64693; abcam, Cambridge, UK) in 0.5% BSA-PBS and incubated overnight at 4°C. Next, slides were incubated with 0.5% BSA-PBS solution with secondary antibodies diluted 1:100 (rabbit-Alexa Fluor647 711-605-152, mouse-Rhodamine (TRITC) AffiniPure 715-025-150; Jackson, ImmunoResearch, Cambridge, UK) and WGA diluted 1:1000 (WGA-Alexa Fluor488 W11261, Molecular probes, Invitrogen) for 1 h at 37 °C. Macrophages in 2D culture were fixed in 4% paraformaldehyde (PFA) for 15 min and permeabilizated with PBS + 0.3% TRITON X-100 (Sigma-Aldrich) for 15 min. Then, the samples were blocked in 3% BSA-PBS for 30 min and incubated with primary human antibody diluted 1:200 (HSPG2) or 1:50 (CD163, BM404; Novus Biological, Centennial, Colorado, USA) in 0.1% BSA-PBS for 1 h at 37 °C. The secondary antibody was diluted 1:200 in 0.1% BSA-PBS (mouse-Rhodamine TRITC) for 1 h at RT. 

Macrophages in t3D culture were fixed in 4% PFA for 3 h at RT, permeabilized with PBS + 0.3% TRITON X-100 for 1:30 h, and incubated in 5% BSA-PBS for 1 h at RT. The 3D constructs were incubated with primary human HSPG2 and CD206 antibodies diluted 1:50 in 1% BSA-PBS overnight at 4°C. Then, 3D samples were incubated with secondary antibodies diluted 1:100 in 1% BSA-PBS (mouse-Rhodamine (TRITC), rabbit–Fluorescein (FITC) AffiniPure 711-095-152, Jackson ImmunoResearch) for 3 h at RT. 

Thereafter, nuclei were counterstained with DAPI (1:1000, Sigma-Aldrich) for 5 min for tissue sections and 2D culture, and for 1 h at RT for t3D culture.

### 4.6. Image Acquisition and Processing

Confocal images of 2D cultures and tissue sections were sequentially acquired by Olympus’ PLAPON 60× OSC2 super-corrected objective confocal apparatus. Sequential 0.5 μm thick Z-stacked sections were imaged through the entire sample using a 60× (for 2D culture) or 20× (for tissue sections) objective lens and were used to create maximum intensity projections (MIPs) and processed with Fiji (National Insitute of Health). All images within given experiment were acquired with the same laser intensity before analysis in Fiji.

For the whole-sample acquisitions of t3D samples, a high-resolution custom-assembled spinning disk confocal microscope was used, employing a Nikon Ti inverted microscope equipped with a Crest Optics X-light V2/VCS scan-head connected to Andor DU888 EMCCD camera, used in 180 micron pinhole configuration. Imaging was performed both at 4× and 10× magnification, acquiring Z-stacks with 1-micron Z-step over 138–228 steps, according to different areas of interest. T3D samples were also acquired with a Leica SP5 laser scanning confocal microscope using both 10× and 20× air objectives, implemented with confocal Zoom-in at either 5× or 9×, for 3D stacks and 3D volume rendering imaging respectively, over 245 sections at 1-micron Z-step. Both datasets from spinning disk and laser-scanning acquisition were further processed and elaborated via NIS-Elements v.5.30 software (Leica Microsystem, Wetzlar, Germany) with GA3 module for digital image segmentation and quantification. Binary masks were created on each specific channel in order to evaluate the number of classified segmented objects and to count nuclei, CD206^+^ cells, HSPG2^+^ cells, and combined double-positive CD206^+^HSPG2^+^ cells. For representative images, a 3D volume rendering of Z-stack volumes acquired via SP5 confocal at different magnifications, and zooming-in (10×/20× + 3× or 5× zoom-in) was performed and elaborated with Fiji (ImageJ) first, then NIS-Elements v.5.30 for Richardson–Lucy deconvolution and 3D rendering.

### 4.7. Co-Immunoprecipitation Assay

Macrophages were washed twice in PBS and harvested by scraping in nuclei lysis buffer (50 mM Tris-HCl pH 8.1, 10 mM EDTA, 1% SDS) supplemented with 1 × Protease and Phosphatase Inhibitor Cocktail (PIC) (Sigma-Aldrich). Cells were then collected and lysed for 10 min at 4 °C after trituration (5×) through a 22 G needle, and then centrifugated for 15 min at 12,000× *g* at 4 °C. Supernatants were collected and pre-cleared with paramagnetic protein G beads (Invitrogen) for 2 h on a rotating platform at 4 °C.

We conserved 1% of total volume as INPUT, while collected samples were incubated on a rotating platform at 4 °C overnight with the following primary antibodies: p65 (D14E12; Cell Signaling Technology, Danvers, MA, USA) and normal rabbit IgGs (12-370; Millipore Sigma) at a concentration of 1µg per 1 × 10^6^ cells. Thereafter, samples were conjugated to paramagnetic protein G beads for 4 h on a rotating platform at 4°C, and immunoprecipitated samples were washed three times in PBS for 10 min at 4°C. Collected beads were resuspended in Laemli Buffer 2× (65.8 mM Tris-HCl pH 6.8, 26.3% (*w*/*v*) glycerol, 2.1% SDS, 0.01% bromophenol blue, 355 mM 2-mercaptoethanol) and heated 10 min at 100 °C; then, p65 immunoprecipitated protein was analyzed for Western blot analyses to detect the co-binding of p52 protein. 

### 4.8. Nuclear-Cytoplasmic Fractionation

Macrophages were washed twice in PBS and harvested by scraping in PBS supplemented with PIC. Cells were then collected as a subcellular fraction and total fraction (1.5% of total volume) and centrifugated 5 min at 1500× *g*, at 4 °C. The total fraction was lysed in RIPA buffer (50 mM Tris-HCl pH 7.5, 150 mM NaCl, 1% NP-40, 0.5% sodium deoxycholate, 0.1% SDS, 1 mM EDTA + PIC) for 30 min in ice and centrifugated for 15 min at 12,000× *g* at 4 °C; then, the supernatant was collected as a total proteins fraction. The subcellular fraction was lysed in buffer A (10 mM HEPES pH 7.9, 10 mM KCl, 0.1 mM EDTA, 0.1 mM EGTA + PIC) for 30 min in ice; then, 0.6% of NP-40 (Sigma-Aldrich) was added for 10 min, and the mixture was centrifugated for 5 min at 3000× *g*, at 4 °C. The supernatant was collected as a cytoplasmic fraction, and then the pellet containing the nuclei was washed three times with PBS, resuspended in an equal amount of buffer B (20 mM HEPES pH 7.9, 400 mM NaCl, 25% glycerol, 0.1 mM EDTA, 0.1 mM EGTA + PIC), and kept overnight at −80 °C. Thereafter, the sample was sonicated twice (MicroTip 1/8” 630-0418–Amplitude 0.5%) for 10 s and centrifugated for 2 min at 10,000× *g* at 4 °C, and following this, the supernatant was collected as the nuclear fraction.

### 4.9. SDS-PAGE and Western Blots

For Western blot analyses, an equal volume of nuclear/cytoplasmic fractions or all the immunoprecipitated proteins by Co-IP were analyzed by using a polyacrylamide gel electrophoresis separation (Bio-Rad, Hercules, Californa, USA) and transferred onto nitrocellulose membranes (Amersham Biosciences, Amersham, UK). After saturation with TBS 0.01% (*v*/*v*) Tween-20 (Sigma-Aldrich) with 5% (*w*/*v*) non-fat dry milk (Applichem, Ottoweg, Darmstadt, Germany) for 1 h at RT, blots were incubated with anti-p65 (1:1000 Cell Signaling Technology), anti-p52 (1:500, 4882; Cell Signaling Technology, for nuclear-fractionation protocol), anti-p52 (1:100, sc-7386; Santa Cruz Biotechnology, Dallas, Texas, USA, for Co-IP analysis), anti-H3 (1:8000, D1H2; Cell Signaling Technology), and anti-GAPDH (1:4000, D16H11; Cell Signaling Technology) overnight at 4 °C. Then, the membranes were incubated for 1 h with HRP-conjugated anti-mouse (1:10,000, 315-035-003; Jackson, ImmunoResearch) or anti-rabbit (1:5000, 111-035-144; Jackson, ImmunoResearch), and chemiluminescence was detected using the ECL Advance Western Blotting Detection Kit (Amersham Biosciences) by ChemiDoc (Bio-Rad). GAPDH and H3 were used for protein normalization of the cytoplasmic and nuclear fractions. Densitometric analyses were performed with the ImageJ software.

### 4.10. Statistical Analysis

Statistical analysis was carried out using Prism 9 (GraphPad Software, La Jolla, California, USA). Data are presented as mean ± standard error of mean (SEM). The statistical significance was assessed by the one-way ANOVA and Tukey correction to evaluate the differences between means, while Student’s t-test was used to analyze the efficacy of QNZ treatment for each gene, compared to control treatment, and for the quantitative analysis of Mφ and M2 macrophages in the t3D samples. *p*-values < 0.033 (*), 0.002 (**), 0.0002 (***), 0.0001 (****) were considered statistically significant. 

## Figures and Tables

**Figure 1 ijms-23-07902-f001:**
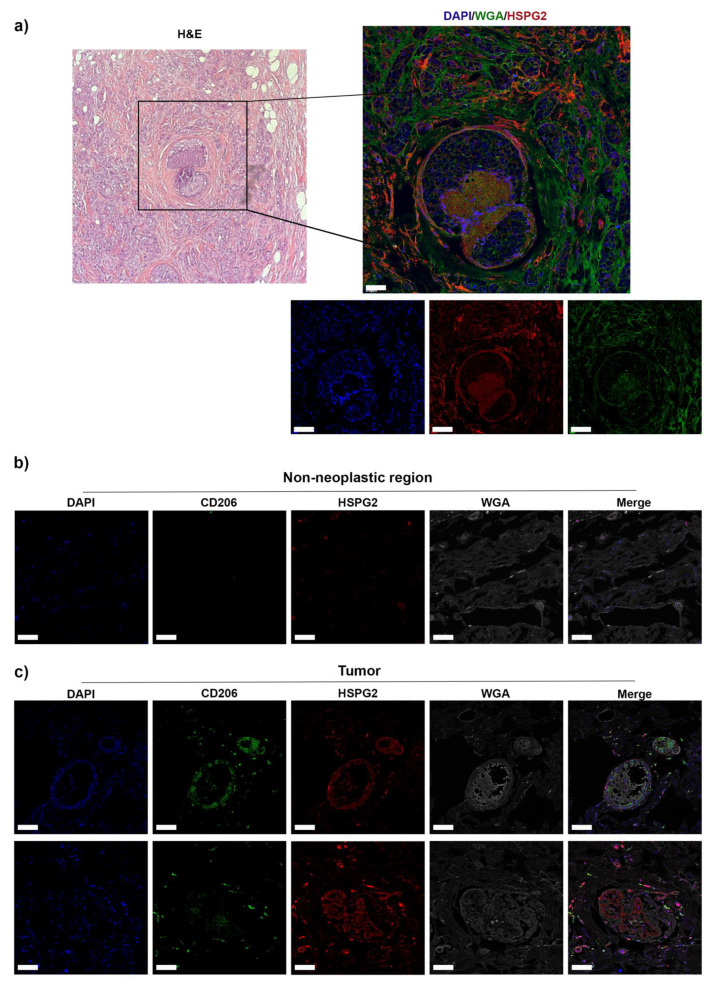
Human triple-negative breast cancer tissue analysis. (**a**) Left panel: Immunohistochemical analysis showing HE staining in malignant tissues was used to identify high cellular density areas and cancer islets. Right panel: Immunofluorescence analysis on the same section was performed to stain HSPG2 in red, and the tissue section was counterstained in green with WGA. Immunofluorescence analysis on sections of tissues derived from the same patients on (**b**) non-neoplastic surrounding tissue and (**c**) tumoral tissue were performed to stain HSPG2 in red and CD206^+^ macrophages in green. Nuclei were counterstained with DAPI, and the tissue section in gray with WGA. Scale bars represent 40 μm. The representative images derived from qualitative analysis of TNBC biopsies of *n* = 3 patients.

**Figure 2 ijms-23-07902-f002:**
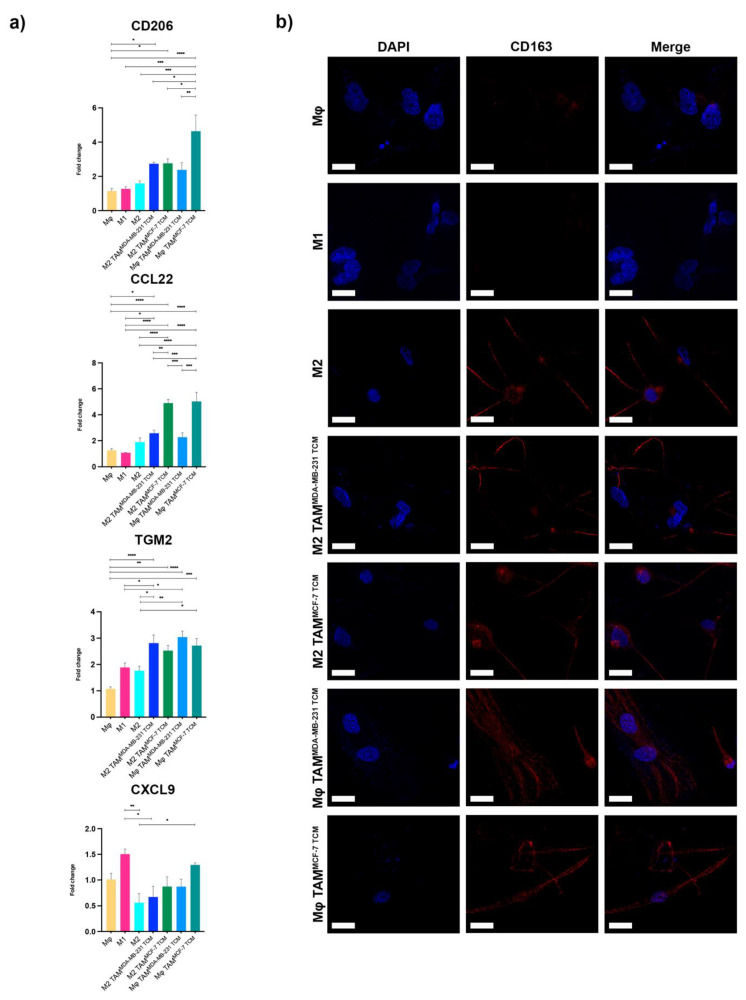
Macrophage marker expression. (**a**) Gene expression analysis of M2-markers evaluated by qRT-PCR. Error bars represent ±SEM. One-way ANOVA and Tukey correction was used to evaluate the differences between means; * *p* < 0.033, ** *p* < 0.002, *** *p* < 0.0002, **** *p* < 0.0001. *n* = 4 for each experimental group. (**b**) Immunofluorescence analysis of the surface marker CD163 showing CD163^+^ macrophages in red. Mφ untreated macrophages and M1-polarized macrophages with LPS were used as negative controls. Nuclei were counterstained with DAPI. Scale bars represent 40 μm.

**Figure 3 ijms-23-07902-f003:**
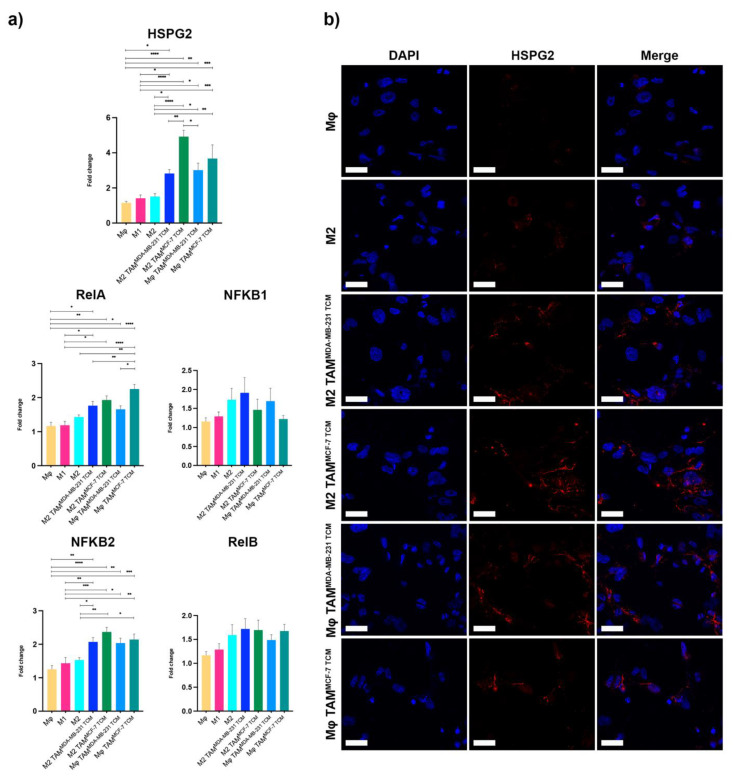
HSPG2 expression in macrophages. (**a**) Gene expression analysis of HSPG2 and canonical and non-canonical NF-κB pathways were evaluated in untreated and treated macrophages by qRT-PCR. Error bars represent ±SEM. One-way ANOVA and Tukey correction were used to evaluate the differences between means; * *p* < 0.033, ** *p* < 0.002, *** *p* < 0.0002, **** *p* < 0.0001. *n* = 4 for each experimental group. (**b**) Immunofluorescence analysis of HSPG2 in untreated and treated macrophages showing HSPG2 in red and nuclei counterstained with DAPI. Scale bars represent 40 μm.

**Figure 4 ijms-23-07902-f004:**
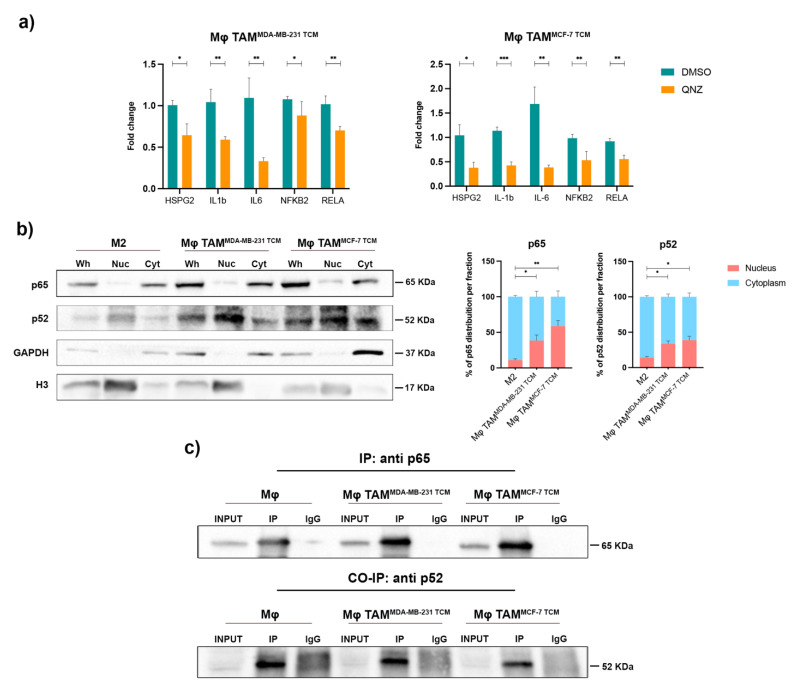
p65/p52 unusual association. (**a**) Gene expression analysis of NF-κB target genes and HSPG2 by qRT-PCR in TAMs treated with QNZ, compared to control DMSO treatment. *n* = 5 for each experimental group. (**b**) Western blot analysis of cytoplasmic and nuclear fractions of treated macrophages. GAPDH and H3 were used as cytoplasmic and nuclear markers (Wh = whole extract; Cyt = cytoplasmic fraction; Nuc = nuclear fraction). *n* = 3 for each experimental group. (**c**) Co-immunoprecipitation (Co-IP) of p65 and p52 proteins in nuclear extracts of untreated and treated macrophages. Upper panel: IP with p65 antibody; lower panel: immunoblot with p52 antibody on the same membrane shown in the upper panel. IP lane was compared to the control IgG lane, INPUT represents 1% of the total nuclear lysate. *n* = 3 for each experimental group. Error bars represent ±SEM. Student’s t-test was used to analyze the efficacy of QNZ treatment for each gene, compared to the control treatment. One-way ANOVA and Tukey correction were used to evaluate the differences between means for the nuclear protein enrichment; * *p* < 0.033, ** *p* < 0.002, *** *p* < 0.0002.

**Figure 5 ijms-23-07902-f005:**
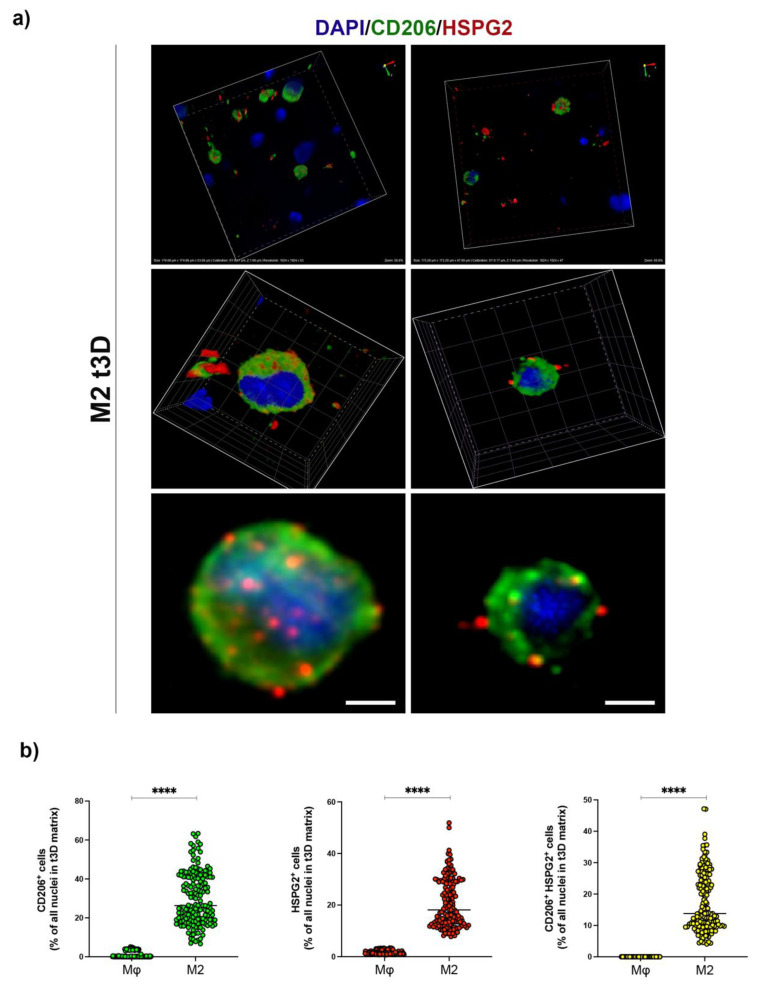
HSPG2 expression in t3D matrix. (**a**) 3D volume rendering of Z-stack volumes acquired at different zooming-in (+3× or 5× zoom-in). The immunofluorescence representative images display M2 embedded in t3D matrix. CD206 was detected in green, HSPG2 in red; nuclei were counterstained with DAPI. Scale bars represent 10 μm. (**b**) Dot-plot graphs showing total quantifications, displayed as percentages of all counted nuclei, of either CD206+ cells (green dots), HSPG2+ cells (red dots), or double-positive CD206+HSPG2+ cells (yellow dots). The analyses were performed on all acquired Z-plans from both Mφ (*n* = 2230) and M2 (*n* = 1114) macrophages in t3D matrix. Error bars represent ±SEM. Student’s t-test; **** *p* < 0.0001.

**Figure 6 ijms-23-07902-f006:**
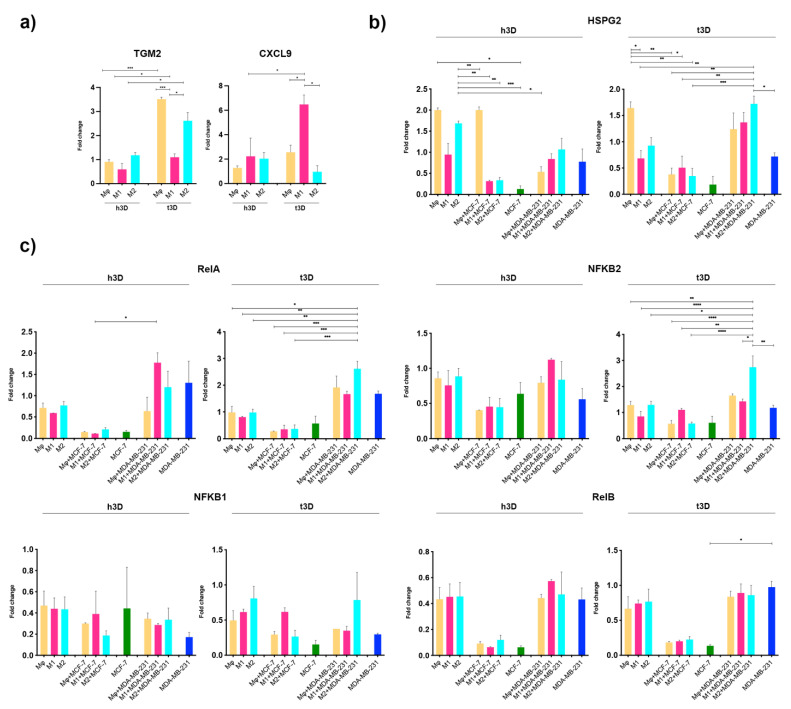
Macrophage genes expression in h3D and t3D matrix. (**a**) M2-markers and for (**b**) HSPG2, (**c**) canonical and non-canonical NF-κB pathway by qRT-PCR. One-way ANOVA and Tukey correction were used to evaluate the differences between means; * *p* < 0.033, ** *p* < 0.002, *** *p* < 0.0002, **** *p* < 0.0001. *n* = 3 for each experimental group.

**Table 1 ijms-23-07902-t001:** Human primer sequences for qRT-PCR analysis.

Gene Symbol	Sense-Forward Primer	Antisense-Reverse Primer
GAPDH	TCTTTTGCGTCGCCAGCCGAG	TGACCAGGCGCCCAATACGAC
HSPG2	TCGGCCATGAGTCCTTCTAC	GTGTATCGCAACTTCCCACC
RelA	ATCCCATCTTTGACAATCGTGC	CTGGTCCCGTGAAATACACCTC
NFKB1	GTGGAGCACGACAAC	GGTGTGGTTCCATCG
NFKB2	TGAAGCCAGTCATCT	CCTGCTGTCTTGTCC
RelB	CATTGAGCGGAAGATTCAAC	GCAGCTCTGATGTGTTTGTG
CD206	CGAGGAAGAGGTTCGGTTCACC	GCAATCCCGGTTCTCATGGC
TGM2	GCAGTGACTTTGACGTCTTTGCCC	GTAGCTGTTGATAACTGGCTCCACG
CCL22	GCCAACATGGAAGACAGCGT	TTATCCCTGAAGGTTAGCAACACCA
CXCL9	GCTGGTTCTGATTGGAGTGC	GAAGGGCTTGGGGCAAATTG
IL-6	TTCTCCACAAGCGCCTTC	AGAGGTGAGTGGCTGTCTGT
IL-1b	CCTGAGCTCGCCAGTGAAA	TCCTGGAAGGAGCACTTCATCT

## Data Availability

All relevant data presented in this study are included in the article and its Appendix A. Any other data that support the findings discussed here are available from the corresponding author upon request.

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
