# Peer review of "Unusual Association of NF-κB Components in Tumor-Associated Macrophages (TAMs) Promotes HSPG2-Mediated Immune-Escaping Mechanism in Breast Cancer"

_ijms, 2022, doi:10.3390/ijms23147902_

Round 1

Reviewer 1 Report

Dear Authors,

I congratulate you on the interesting topic.

In this paper, you provided an in vitro study to better understand the mechanisms that contributes to the chemical and mechanical modification of the environment surrounding the tumor, exasperating immune-escaping mechanism. By evaluating the role of TAMs and NF-kB transcription factors, they have demonstrated the existence of an unusual association of TAMs and NK-kB components that determines the deposition of heparin sulfate proteoglycan 2 (HSPG2), also known as perlecan. HSPG2 is involved in drug resistance and it actively contributes to support tumor growth by sequestering proteins growth factors.

In Triple Negative Breast Cancers (TNBC) the tumor microenvironment (TME) exhibits a distinctive stromal deposition of HSPG2, in high cell density areas, and surrounding tumor islands, where TAMs are highly infiltrated, mimicking a physical shield. The manipulation of TME could represent a promising clinical strategy for precision immunotherapy in cancer patients.

In conclusion, this study is very interesting and promising, and it provides relevant data that could be a stimulus towards a more in-depth evaluation of this argument.

The paper is well written, original and within the scope of the journal and it could be accepted for publication in present form.

Kind regards

Reviewer 2 Report

A better understanding of the tumor microenvironment is needed, especially in context of immune evasion. This reviewer has read with interest this study on breast cancer by De Paolis et al, that highlights a new mechanism.

The experimental work is sound, and it demonstrates the unusual association of NF-kB in tumor-associated macrophages in the deposition of HSPG2 which in turn affects the stiffness of the tumoral extracellular matrix. The manuscript is well written.

Minor comments

l  Minor spelling checks and linguistic polishing is needed

l  Please clearly indicates where the tissue samples came from. Please provide more information about these samples.
